# Arginine Hydrochloride Reduce Rectal Mucosal Irritation of Sodium Aescinate: Molecular Docking, Physical Properties, Anti-Hemorrhoidal Activity, Safety and Topical Gel Formulations Investigation

**DOI:** 10.3390/pharmaceutics16121498

**Published:** 2024-11-22

**Authors:** Di Hu, Qiuyang Zeng, Huanrong Wang, Wei Jiang

**Affiliations:** School of Life Science and Technology, Wuhan Polytechnic University, Wuhan 430023, China; 20210212022@whpu.edu.cn (D.H.);

**Keywords:** sodium aescinate, arginine hydrochloride, rectal irritation, inflammation, hemorrhoid, gels

## Abstract

**Background/Objectives**: Sodium aescinate (SA) is commonly used topically due to its anti-inflammatory, anti-edematous, and anti-swelling properties. However, the clinical application of SA is limited by strong irritation, and cannot be used on the damaged skin and mucous membrane. This study aimed to investigate whether arginine hydrochloride (Arg·HCl) could reduce the rectal mucosal irritation of SA through the formation of a gel. **Methods**: Molecular docking was first used to explore potential interactions between SA and Arg·HCl. Gels for rectal administration were then formulated by combining SA with various ratios of Arg·HCl (from 1:0 to 1:10). In vitro tests, including pH, centrifuge stability, viscosity, and spreadability analysis, were conducted. The optimal gel formulation was determined based on rectal mucosal irritation tests and anti-inflammatory experiments. Additionally, the anti-hemorrhoidal characteristics and safety of the optimal gel in terms of acute toxicity and dermal sensitivity were evaluated. **Results**: The optimal SA to Arg·HCl ratio of 1:6 (F5-SA gel) was identified, significantly reducing rectal mucosal irritation while enhancing anti-inflammatory activity. The F5-SA gel demonstrated high efficacy against hemorrhoids, notably promoting anal ulcer healing. When administered rectally to rabbits at a dose of 132 mg·kg^−1^·d^−1^ (198 times the recommended therapeutic dose), no other obvious side effects were observed except a significant reduction in food intake on the day of administration. In addition, the gel did not induce dermal sensitivity. **Conclusions**: The F5-SA gel is a promising formulation that can reduce irritation and toxic side effects, and enhance the therapeutic effect to some extent, ultimately achieving a safer and more effective rectal delivery system for SA.

## 1. Introduction

Sodium aescinate (SA), derived from horse chestnut (*Aesculus hippocastanum*) seed extracts, serves as a key active compound in this traditional medicinal herb [1]. It appears as a white powder or crystalline substance, comprising a mixture of triterpene saponins, including SA A, B, C, and D (Figure 1) [2]. Extensive research has highlighted the potent anti-inflammatory and anti-edematous properties of SA, making it a widely utilized treatment in clinical practice [3]. Its applications include managing hemorrhoids, chronic venous insufficiency, encephaledema, and swelling resulting from trauma or surgery [4,5]. SA is available in various formulations, such as oral tablets, injections, and topical gels [6]. Among these, external applications like SA liniment or compound SA gel have demonstrated effective results for ecchymosis, sprains, and acute soft tissue injuries. However, certain limitations are associated with these external preparations, including restrictions on use over damaged skin and mucous membranes [7]. Furthermore, rectal administration of SA, while potentially effective for hemorrhoids and postoperative recovery, may cause irritation and discomfort, leading to low patient compliance.

Amino acids are essential for the synthesis of proteins, which play a key role in the repair and growth of tissues such as muscles, skin, and mucosa. Certain amino acids, such as arginine, have important roles in regulating the immune system and maintaining the barrier function of gastrointestinal mucosa [8]. Amino acids have been multifunctional excipients for improving tissue tolerability for many therapeutic classes. It is well documented that specific essential amino acids can substantially alleviate aspirin-induced damage to the gastric mucosa [9,10,11]. According to Lim et al. [12], methionine and histidine significantly reduced the extent of gastric mucosal injury resulting from oral aspirin administration. Yamazaki et al. [13] also reported that rectal mucosa irritation of diclofenac (DC) could be reduced by their administration in the form of salts of basic amino acids with favorable effects on bioavailability. The irritation of DC salt with L-arginine (DC-Arg) showed the weakest effects on the rectal mucosa.

To our knowledge, no attempt has been made yet to administer SA rectally for assessing hemorrhoid healing activity. Following molecular docking studies of SA with arginine hydrochloride (Arg·HCl), we initiated research into the use of Arg·HCl in rectal delivery formulations of SA, aiming to minimize potential mucosal damage at the application site. To mitigate the mucosal irritation caused by SA, six different weight ratios of SA to Arg·HCl were tested, ranging from 1:0 to 1:10. These formulations were developed as gels and subjected to rectal mucosal irritation tests and anti-inflammatory activity to identify the optimal ratio of SA to Arg·HCl. Their physical properties in vitro were evaluated. Moreover, with the most effective formula, the anti-hemorrhoidal activity and safety characteristics of SA gel have been investigated by animal models in vivo.

## 2. Materials and Methods

### 2.1. Materials

Materials used in this study included SA (Wuxi Kaifu Pharmaceutical Co., Ltd., Wuxi, China), glycerinum, propylene glycol, and edetate disodium (Hunan Er-kang Pharmaceutical Co., Ltd., Changsha, China), ethylparaben (Taishan Xinning Pharmaceutical Co., Ltd., Taishan, China), carboxymethylcellulose sodium (CMC-Na) and hydroxy propyl methyl cellulose (HPMC) (Anhui Sunhere Pharmaceutical Excipients Co., Ltd., Huainan, China), arginine hydrochloride (Shanghai Xiehe Amino Acid Co., Ltd., Shanghai, China). All other chemicals were of analytical grade. 

### 2.2. Animals

Kunming mice (18–22 g), SD rats (180–250 g), Japanese big-ear rabbits (2.0–2.5 kg), and Hartley guinea pigs (200–279 g) purchased from Hubei Provincial Laboratory Animal Research Center (Wuhan, China). The animals were kept at 25 ± 2 °C with a relative humidity of 50–60% in the experimental animal center of Wuhan Polytechnic University. According to the Guide for the Ethical Care and Use of Laboratory Animals, these studies have been approved for animal studies by the Ethical Committee of Wuhan Polytechnic University.

### 2.3. Formulation of Gel

Carefully weighed SA and the excipients, including Arg·HCl, CMC-Na, HPMC, glycerol, propylene glycol, disodium edetate, and ethylparaben. The specific process involved adding CMC-Na and HPMC to an appropriate amount of distilled water and letting it sit overnight until fully swollen. Dissolve the accurately weighed SA, Arg·HCl, glycerol, propylene glycol, disodium edetate, and ethylparaben in a certain amount of distilled water, then add the solution to the gel matrix. Well mixed. The procedure of the gel formulation is shown in Figure 2. Six formulations (F1~F6) were prepared, each containing 2% SA mixed with varying amounts of Arg·HCl (SA:Arg·HCl = 1:0, 1:0.5, 1:1, 1:3, 1:6, and 1:10) (Table 1).

### 2.4. Molecular Docking

Molecular docking was conducted using AutoDock Vina to evaluate the binding affinities of SA. The potential binding interactions between SA and Arg·HCl were visualized, and the binding constants were estimated using Pymol for further analysis.

### 2.5. Property Assessment of the SA Gels

The physical characteristics of various SA gel formulations (F1~F6) were examined through a series of standardized tests, including organoleptic evaluation, pH measurement, viscosity analysis, spreadability assessment, and homogeneity checks.

#### 2.5.1. Organoleptic Evaluation

The organoleptic properties, such as color, consistency, homogeneity, phase stability, and texture, were assessed 24 h after gel preparation. To assess color, a thin layer of the gel was spread on white paper to maximize contrast, allowing for a clearer comparison with the base gels [14].

#### 2.5.2. pH Measurement

The pH of each formulation was measured using a digital pH meter (Mettler Toledo, Columbus, OH, USA). To perform the test, 2 g of gel was diluted in 10 mL of distilled water and assessed using a standard buffer solution that had been calibrated at a temperature of 25 ± 2 °C. Each measurement was repeated three times to ensure accuracy.

#### 2.5.3. Centrifugation Stability Test

The stability of the gel samples was assessed by subjecting them to centrifugation at 3000 rpm for 30 min using a laboratory centrifuge (Hettich Mikro 120, Tuttlingen, Germany) maintained at room temperature (25 ± 2 °C). After centrifugation, the samples were visually examined for any indications of phase separation or physical instability [15].

#### 2.5.4. Spreadability Test

The spreadability of each gel formulation was tested by placing 0.5 g of gel within a circular area of 1 cm diameter on a glass plate. A second glass plate was placed over it, and a weight of 100 g was applied on top. The time required for the two plates to separate was recorded, and the spreadability was determined using a specific calculation Formula (1) [16].
S = M × L/T(1)

M: weight tied on upper slide; L: length of glass slide; T: time in s.

#### 2.5.5. Viscosity Test

The viscosity of each gel formulation was analyzed using a viscometer (Brookfield DV2TRV, Middleboro, MA, USA) to assess the rheological properties. A spindle number 4 was used for the test, operated at a rotational speed of 10 rpm under controlled conditions at 30 °C.

### 2.6. Rectal Irritation Test

Forty-eight rabbits (half male and half female) were randomly divided into eight groups (six rabbits in each group): a normal saline (NS) group, a gel matrix control group (without SA and Arg·HCl), and six groups with different ratios of SA and Arg·HCl (F1~F6, respectively). Seven days after rectal administration of the gel, four rabbits from each group were sacrificed, and the rectum was isolated. Observations were made to assess whether the mucous membrane exhibited hyperemia, swelling, or secretion. Additionally, the rectum was prepared for histopathological examination by rinsing it with saline and storing it in 10% v/v formalin solution. Macroscopic examination of the specimens was conducted before cutting them longitudinally into three sections. The samples were then dehydrated, embedded, sectioned, and stained with hematoxylin and eosin (H&E). The tissues were observed under a microscope for signs of lesions, including vascular congestion, mucosal edema, leukocyte infiltration, and epithelial ulcers. The remaining two rabbits in each group were monitored daily. Seven days after discontinuation of the drug, they were euthanized to assess for any signs of rectal irritation [17].

### 2.7. Aiti-Inflammatory Activity

#### 2.7.1. Xylene-Induced Ear Swelling in Mice

Ninety Kunming mice were equally randomized into nine groups: a blank control group (NS), a gel matrix control group, a positive drug group (Dexamethasone acetate ointment (10%), 40 mg·kg^−1^ of body weight), and six groups with different ratios of SA and Arg·HCl (F1~F6, respectively, 20 mg·kg^−1^). For three consecutive days, the test compound was administered to the mice. Thirty minutes after the final administration, 0.03 mL of xylene was applied to the right ear of each mouse to induce inflammation, while the left ear, without xylene treatment, served as a control. Thirty minutes after the induction of inflammation, the mice were euthanized. Using an 8 mm perforator, identical sections were excised from both ears along the auricle baseline. These sections were then weighed on an electronic balance. The difference in weight (mg) between the right and left ear samples was recorded as the degree of swelling. The inhibition rate of swelling was subsequently calculated based on these measurements [18].

#### 2.7.2. Histamine-Induced Capillary Permeability in Rats

The grouping and administration for this assay followed the same protocol as described in the xylene-induced ear swelling test. After removing hair from a 1.5 cm × 1.5 cm area using an 8% sodium sulfide solution, a vascular permeability test was initiated by subcutaneous injection of 0.05 mL of freshly prepared 0.01% histamine phosphate in 0.9% normal saline. Subsequently, the rats were treated with the respective topical formulations. Following this, 0.05 mL of 1% Evans blue dye in saline was injected intravenously via the tail vein. Thirty minutes after the dye injection, the rats were euthanized. Blue-stained tissue samples were excised immediately and immersed in an acetone-saline solution (7:3) for 48 h. The optical density (OD) of the supernatant was measured at 610 nm to determine the absorbance of Evans blue. This absorbance was used as an indicator of capillary permeability in the tissue [18].

### 2.8. Anti-Hemorrhoidal Activity

#### 2.8.1. Croton Oil Caused Anus Swelling in Rats

Male rats were randomly assigned to five groups, each consisting of 10 animals: an F5-SA gel matrix control group, a positive control group treated with Mayinglong musk hemorrhoids ointment, and groups treated with various concentrations of F5-SA gel (10 mg·kg^−1^, 20 mg·kg^−1^, and 40 mg·kg^−1^). Hemorrhoids were induced by gently rubbing the anorectal area with a sterile cotton swab for 10 s daily. The swab was moistened with a mixture prepared from croton oil (6% in diethyl ether) combined with deionized water, pyridine, and diethyl ether in a ratio of 1:4:5:10. Five days after inducing hemorrhoids with croton oil, the rats were treated for seven consecutive days with the respective treatments for each group. Thirty minutes after the final administration, all rats were euthanized. Rectal tissue was excised 15 mm from the anal margin, rinsed with cold saline, and blotted dry with filter paper. The rats’ anorectal histology samples were used to assess anal swelling in each group [19,20]. The therapeutic effects of F5-SA gel were evaluated using the rectoanal swelling coefficient as a key indicator. The rectoanal swelling coefficient was calculated using Equation (2)
(2)$$\mathrm{Rectoanal}\;\mathrm{swelling}\;\mathrm{coefficient}\;(\%)=\frac{Ww-Wd}{Ww}\times 100$$where W_w_ is the weight of rectoanal wet tissue and W_d_ is the weight of rectoanal dry tissue.

#### 2.8.2. Acetic Acid-Induced Anal Ulcer in Rabbits

The animals were randomly divided into 5 groups with 10 animals in each one: an F5-SA gel matrix control group, a positive control group treated with Mayinglong musk hemorrhoids ointment, and groups treated with various concentrations of F5-SA gel (2 mg·kg^−1^, 4 mg·kg^−1^, and 8 mg·kg^−1^). After a 24 h fasting period, rabbits were anesthetized via intraperitoneal injection of 7% chloral hydrate (0.5 mL per 100 g body weight). To establish the anal ulcer model, 20 μL of 36% glacial acetic acid was injected subcutaneously at four points around the anal area. Within approximately 24 h, observable anal ulcers characterized by redness and inflammatory exudation developed [21]. Treatments were administered once daily for a period of seven consecutive days to evaluate therapeutic efficacy. On the 8th day, the ulcer healing status was observed and recorded, and the severity of the ulcers in each group was assessed according to the following scoring criteria:1 point: ulcer bleeding (++) and significant ulcer exudate (+++);2 points: moderate ulcer exudate (++) and slight ulcer bleeding (+);3 points: slight ulcer exudate (+);4 points: scab formation, indicating near-complete healing.

The ulcer healing score was used as the indicator to evaluate the therapeutic effect of F5-SA gel on acetic acid-induced anal ulcers in rabbits.

### 2.9. Acute Toxicity

Rabbits were divided into two groups, each consisting of 6 animals: group I (control, treated with F5-SA gel matrix) and group II (treated with F5-SA gel, 2.2 g·kg^−1^·d^−1^, twice, containing the highest concentration of SA at 6%). After fasting for approximately 16 h, the rabbits were secured in a rabbit restrainer with their hindquarters elevated at an angle of about 45 degrees. Using a syringe, the medication was administered into the rectum at a dose of 1 mL/kg per application, ensuring the medication remained in the rectum for 4 h (2 h in the morning and 2 h in the afternoon), with one administration in the morning and one in the afternoon. Toxicity in the rabbits was assessed over a 7-day treatment period followed by a one-week recovery phase. The maximum administration method was adopted, involving the highest concentration and volume of the gel. Mortality and general health were monitored at least once a day [22]. Body weight was recorded on days 1, 2, 5, 7, 10, and 14, while food intake was measured on days 0, 1, 2, 5, 7, 10, and 14 to assess the impact of the treatment on their nutritional intake and overall well-being. 

### 2.10. Dermal Sensitivity Studies

The Magnussen and Kligman guinea pig maximization test [23] was used to test F5-SA gel for its capacity to sensitize skin for allergic contact sensitivity. Thirty guinea pigs were randomly divided into a matrix control group (F5-SA gel matrix, 0.5 g), a positive control group (2,4-dinitrochlorobenzene, DNCB; sensitization concentration 1%, elicitation concentration 0.1%, 0.5 mL), and a treatment group (F5-SA gel, sensitization concentration 4%, elicitation concentration 2%, 0.5 g), with 10 animals in each group, half male and half female. Prior to the experiment, the skin on both sides of the back of the test animals was shaved, covering an area of approximately 3 cm × 3 cm. The corresponding substances (F5-SA gel matrix, 2,4-dinitrochlorobenzene, and F5-SA gel) were applied to the left shaved area of each group, and secured with non-irritating gauze and tape for 6 h. Sensitization occurred on days 1, 7, and 14. Fourteen days after the final sensitization, the same substances, at their elicitation concentrations, were applied to the right shave for 6 h. Allergic reactions, including erythema and edema, were observed at 1, 24, 48, and 72 h post-application. The scoring criteria were as follows: erythema, 0 (negative), 1 (mild), 2 (moderate), 3 (severe), and 4 (edematous erythema); edema, 0 (negative), 1 (mild), 2 (moderate), and 3 (severe).

The Kligman scheme of classification [23] was used to assign a grade of sensitization of I-V according to the percentage of animals in the test group sensitized as follows: ≤8% animals sensitized, weak, grade I; ≥9% to 28%, mild, grade II; ≥29% to 64%, moderate, grade II; ≥65% to 80%, strong, grade IV; and ≥80% to 100%, extreme, grade V.

### 2.11. Statistical Analysis

The data were analyzed statistically using GraphPad Prism software (version 9, San Diego, CA, USA). Results are expressed as the mean ± standard deviation (SD). A one-way ANOVA was performed to evaluate differences among multiple unpaired groups, while a Student’s *t*-test was applied for comparisons between two unpaired groups. Statistical significance was denoted as * *p* < 0.05 (moderately significant) and ** *p* < 0.01 (highly significant).

## 3. Results and Discussion

### 3.1. Molecular Docking Analysis

The potential interaction mode between SA and Arg·HCl was investigated through a molecular docking study. The results showed that the hydroxyl and the hydroxymethyl groups in SA could be possible interaction sites with Arg·HCl, where the terminal guanidine group could form an H-bond with a binding affinity of −3.9 kcal/mol (Figure 3). This potential interaction indicates they may form multicomponent crystals, where two or more molecules are held together by intermolecular forces, creating a stable crystalline structure [24]. Furthermore, Arg·HCl is amphiprotic, meaning it can both donate and accept protons and has the ability to co-crystallize with various guest molecules. When the difference between the pKa values of the base and acid is less than 3, it is typically indicative of the formation of a salt-cocrystal continuum [24]. In the present case, with the pKa of SA (base) being approximately 7.3 and the pKa of Arg·HCl (acid) ranging from 5.5 to 7.0, this strongly supports the formation of a soluble salt-cocrystal continuum when both components are dissolved together. The theoretical calculations implemented supported a salt-cocrystal continuum formation between Arg·HCl and SA, explaining that the irritability of SA was reduced.

### 3.2. Physical Properties Evaluation of the SA Gels

All the SA Gels (F1~F6) exhibited a colorless to light yellow appearance, a smooth and non-greasy texture, and a shiny finish (Figure 4). They were easy to wash off and displayed a homogeneous composition with a uniform and satisfactory consistency.

#### 3.2.1. pH

Gels are commonly applied to the skin or mucous membranes, where the natural pH typically ranges from 4.5 to 7.0. To prevent irritation or disruption of the skin barrier, it is essential that the pH of the gel aligns closely with this natural range [25]. Therefore, careful consideration of the pH and buffering capacity of topical products is essential to ensure safety and compatibility with the skin. In this study, the pH values of all gel formulations ranged from 5.85 to 7.00 (Table 2), which aligns closely with the natural pH of the skin and body cavities. These results suggest that the gels are suitable for topical application and are unlikely to cause irritation or other adverse skin reactions.

#### 3.2.2. Centrifugation

Centrifugation is a well-established technique for detecting potential instabilities in pharmaceutical formulations, particularly for gels. By subjecting these formulations to stress conditions that simulate elevated gravitational forces and increased particle mobility, centrifugation helps identify issues such as decomposition, phase separation, and sedimentation. This method provides critical insights into the formulation’s stability and indicates whether reformulation may be necessary [26,27]. In our study, none of the formulations (F1~F6) exhibited any signs of decomposition, separation, or precipitation following centrifugation.

#### 3.2.3. Viscosity

Proper viscosity is crucial as it ensures that the gel remains at the application site for an adequate duration [28]. The viscosity of the gel formulations ranged from 378.4 to 422.5 cps. A positive correlation was noted between the viscosity and the concentration of Arg·HCl, as detailed in Table 2. This indicates that higher concentrations of Arg·HCl contributed to an increase in the gel’s thickness and consistency.

#### 3.2.4. Spreadability

The spreadability of the gel formulations was measured and found to range between 3.30 and 4.37 cm, as shown in Table 2. Spreadability is closely linked to structural viscosity, with lower viscosity leading to enhanced spreadability. The results showed a strong correlation between viscosity and spreadability, emphasizing the formulation’s desirable application characteristics [29].

### 3.3. Exploring the Optimal Arg·HCl Concentration 

#### 3.3.1. Evaluation of F1–F6 SA Gels on Rectal Irritation Test

The findings from the rectal mucosal irritation studies are presented in Figure 5. The NS group and the SA gel matrix group maintained intact mucosal layers with no signs of hyperaemia in the submucosal or serosal layers, indicating no irritation. However, the group treated with the SA gel lacking Arg·HCl (F1) exhibited severe mucosal surface shedding and significant hyperaemia in both the submucosal and serosal layers, pointing to substantial rectal mucosal damage. In the F2 group, secretions were observed in the intestinal cavity, with an intact mucosal layer but the presence of hyperaemia and significant lymphocyte infiltration in the lamina propria, indicating moderate irritation. Similarly, the F3 group displayed an intact mucosal layer but with hyperaemia in both the submucosal and serosal layers, also suggesting moderate rectal mucosal damage. In contrast, the F4 group exhibited only partial mucosal shedding without congestion in the submucosal and serosal layers, indicating mild mucosal irritation. The F5 and F6 groups, however, maintained intact mucosal layers without signs of hyperaemia, with the rectal pathology sections appearing largely normal, demonstrating minimal to no irritation.

Upon examining the rectal mucosa one week after discontinuation of the treatment, only the F1 group continued to show signs of mucosal hyperaemia, while all other groups displayed normal mucosal conditions (Figure 6).

These results clearly show that SA alone causes significant irritation to the rectal mucosa, as observed in the F1 group, compared to the control matrix group. However, rectal mucosal irritation was significantly reduced, following the application of SA gels containing Arg·HCl. Remarkably, as the concentration of Arg·HCl increased, the mucosal irritation caused by SA progressively decreased. When the ratio of Arg·HCl to SA reached 1:6 (F5 and F6), no significant irritant effects, such as hyperaemia, edema, or secretion, were observed. These findings suggest that F5 and F6 formulations did not cause any marked irritation in the rectum of the rabbits.

The rectal mucosa is highly complex and heterogeneous in both histological and cellular terms. The precise mechanism by which SA causes mucous membrane irritation is not fully understood. Arg·HCl, an amino acid with a pKa of 5.5–7.0, likely forms a salt-cocrystal continuum with SA. The delocalization of the positive charge, facilitated by the conjugation between the double bond and the nitrogen atom’s lone pairs, promotes the formation of several hydrogen bonds. These findings are consistent with the results observed in the molecular docking studies. Furthermore, arginine exerts protective effects by restoring disrupted barrier function and maintaining gastrointestinal mucosal integrity. It influences intestinal permeability by regulating the expression of tight junction proteins, such as claudin-1, occludin, and ZO-1, which are crucial for maintaining the integrity of the mucosal barrier [30]. These findings help explain why Arg·HCl effectively reduces the mucosal irritation caused by SA in the tested formulations.

#### 3.3.2. Evaluation of F1~F6 SA Gels on Xylene-Induced Ear Swelling

The xylene-induced ear edema model in mice is a straightforward and effective method for assessing the anti-inflammatory potential of compounds. It serves as an initial screening model to evaluate the effectiveness of potential anti-inflammatory agents in reducing acute inflammation [31]. The results of the in vivo ear swelling assay for the F1~F6 SA gels are presented in Figure 7. Xylene injection led to a significant increase in ear edema, representing acute inflammation, compared to the baseline values. However, following the application of the F1~F6 SA gels, the degree of ear swelling was markedly reduced in comparison to the matrix control group (*p* < 0.05, or *p* < 0.01), demonstrating the anti-inflammatory potential of these formulations. In contrast, the group receiving only the gel matrix and the NS group did not exhibit any significant changes in inflammation, indicating that the matrix alone did not contribute to reducing inflammation. The inhibitory effect on ear edema was significant across all SA gel-treated groups, with inhibition rates of 39.7%, 32.3%, 35.6%, 37.4%, 44.1%, and 34.7% for the F1, F2, F3, F4, F5, and F6 formulation, respectively. These results demonstrated that all SA gel formulations exhibited stronger anti-inflammatory effects than the positive control group treated with dexamethasone, which had an inhibition rate of 29.7%. Notably, the F5 formulation produced the highest inhibition rate of 44.1%, indicating that it was the most effective in reducing xylene-induced ear edema in mice.

The findings from this study clearly suggest that the F5 formulation has the strongest anti-inflammatory activity among the tested SA gels, offering superior efficacy in reducing ear swelling compared to both the positive control and other formulations.

#### 3.3.3. Evaluation of F1~F6 SA Gels on Histamine-Induced Capillary Permeability

The histamine-induced capillary permeability model is widely used to investigate the effects of drugs on vascular permeability and inflammatory responses [32]. In this experiment, the impact of the tested formulations on vascular permeability was measured by optical density at 610 nm (*OD*_610_). As shown in Figure 8, histamine-induced enhancement of vascular permeability was significantly inhibited in rats pretreated with rectally administered with F1, F2, F4, and F5 (*p* < 0.05, or *p* < 0.01, compared to the matrix control group). The inhibition rates were 38.0%, 31.5%, 29.8%, and 32.5% for F1, F2, F4, and F5, respectively. The positive control drug, dexamethasone, resulted in a significant reduction of 28.4% in vascular permeability compared to the matrix control group.

Among the formulations, F1 demonstrated the highest efficacy in suppressing capillary permeability, followed closely by F5, indicating strong anti-inflammatory potential. These results suggest that both F1 and F5 formulations effectively reduce histamine-induced vascular permeability, with F1 showing the greatest overall inhibition.

In the present study, a combination of rectal irritation testing and two classic acute inflammatory animal models (xylene-induced ear edema and histamine-induced capillary permeability) was used to explore the optimal concentration of Arg·HCl in SA gels. The irritation study confirmed that both F5- and F6-SA gels showed no signs of rectal irritation, making them suitable for rectal application. The results from the anti-inflammatory activity studies revealed that F1 and F5 exhibited the highest inhibition rates and the best anti-inflammatory activity in both models.

Based on the combined experimental results, the optimal ratio of SA to Arg·HCl was determined to be 1:6, corresponding to the F5-SA gel formulation. This formulation effectively balanced anti-inflammatory efficacy with minimal irritation, making it the most promising candidate for further in vivo evaluation.

### 3.4. Anti-Hemorrhoidal Activity Investigation

#### 3.4.1. Evaluation of F5-SA Gel on Croton Oil-Caused Anus Swelling in Rats

Croton oil is a commonly used inflammatory agent to induce experimental hemorrhoids, and the procedure for establishing this model is both straightforward and reliable. The anal weight changes observed in this model serve as key indicators of the severity of the pathological condition. Croton oil-induced lesions develop rapidly, persist for an extended period, and are characterized by distinct histological changes, such as inflammation and edema [19]. These characteristics make the model ideal for assessing potential anti-hemorrhoidal treatments.

As shown in Figure 9, both the positive control and the F5-SA gel groups, at medium and high doses, significantly reduced anal swelling and inflammatory responses caused by croton oil in rats. The rectoanal swelling coefficient in these treatment groups was notably lower than that of the matrix control group, with the differences being statistically significant (*p* < 0.05 or *p* < 0.01). These findings suggest that the F5-SA gel effectively mitigated the inflammation and swelling associated with hemorrhoid-like conditions.

The results of the study showed that the F5-SA gel significantly alleviated croton oil-induced anal swelling in rats, particularly at medium and high doses. This suggests that the F5-SA gel not only reduces inflammation but also offers a protective effect against the pathological changes induced by croton oil, making it a promising candidate for further development as an anti-hemorrhoidal treatment.

#### 3.4.2. Evaluation of F5-SA Gel on Acetic Acid-Induced Anal Ulcer in Rabbits

The acetic acid-induced anal ulcer model closely mimics acute hemorrhoidal symptoms, including pain, hematochezia, redness, swelling, exudation, and mucosal erosion. It is ideal for evaluating anti-hemorrhoidal treatments due to its prolonged symptom duration and reproducibility, allowing assessment of therapeutic efficacy in mucosal protection, anti-inflammatory effects, and swelling reduction [33].

At eight days post-treatment, the low, medium, and high-dose F5-SA gel groups showed significant promotion of anal ulcer healing induced by acetic acid in rabbits. The average scores for ulcer healing in these groups were notably higher than those in the matrix control group, with statistically significant differences (*p* < 0.05, *p* < 0.01) (Figure 10). The F5-SA gel treatment markedly reduced redness, swelling, and inflammatory exudation, while accelerating the healing of ulcerative lesions.

This study demonstrated that the F5-SA gel has a potent effect on enhancing the healing of acetic acid-induced anal ulcers, suggesting its efficacy in reducing inflammation, promoting mucosal repair, and alleviating symptoms commonly associated with hemorrhoidal conditions. These findings further support the potential of F5-SA gel as a therapeutic agent for treating hemorrhoids, particularly during acute flare-ups.

### 3.5. Evaluation of F5-SA Gel on Acute Toxicity

The data revealed no significant differences in body weight gain between group I (control) and group II (F5-SA gel) throughout the study period (Table 3), indicating that the treatment did not adversely affect the overall growth of the animals.

On the day of administration, the food intake in the F5-SA gel-treated group was significantly lower compared to the control group, with a statistically significant difference (*p* < 0.05). However, on days 1, 2, 5, 7, 10, and 14 post-administration, there were no significant differences in food intake between the F5-SA gel group and the control group (*p* > 0.05) (Table 4). This suggests that the initial reduction in food intake was transient and did not persist over the course of the study.

Rabbits were given a single rectal administration of F5-SA gel at a dose of 132 mg·kg^−1^·d^−1^, which is 198 times the recommended therapeutic dose. Aside from the significant but temporary reduction in food intake on the day of administration, no other notable side effects or signs of acute toxicity were observed in the treatment group. These findings suggest that the F5-SA gel is well-tolerated, even at a dose much higher than the therapeutic level, with minimal impact on the overall health and behavior of the animals.

### 3.6. Evaluation of F5-SA Gel on Dermal Sensitivity

In the SA gel group (4%), after sensitization on days 0, 7, and 14, the animals’ skin exhibited mild to moderate erythema, with some animals showing slight desquamation. The erythema resolved within 24 h, while the desquamation subsided within 48 h. In the matrix control group, no abnormalities were observed in the skin after sensitization on days 0, 7, and 14. In contrast, in the positive control group (DNCB-treated), 100% of the animals’ skin exhibited mild to severe erythema after sensitization on days 0, 7, and 14, with some animals also developing edema and desquamation.

The body weight of guinea pigs in the F5-SA gel group increased normally, with no significant difference compared to the matrix control group (*p* > 0.05), indicating that the treatment had no adverse effects on overall health or growth (Table 5).

In the matrix control group, no allergic reactions were observed, with a sensitization rate of 0% at 1, 24, 48, and 72 h after provocation. In the DNCB positive control group, all animals exhibited erythema and edema, with sensitization rates of 100% at 1, 24, 48, and 72 h post-provocation. The average reaction scores were 3.8, 2.8, 1.5, and 0.6, respectively, indicating a strong sensitization response. In the F5-SA gel group (2%), no allergic reactions were observed, and the sensitization rate remained at 0% across all time points, demonstrating that the F5-SA gel does not cause dermal sensitivity. The experimental results are summarized in Table 6.

Our study revealed that F5-SA gel (2%) did not produce any signs of dermal sensitivity, such as erythema, edema, or other allergic reactions, indicating that it is safe for topical use without the risk of sensitization.

## 4. Conclusions

This study successfully developed an optimized gel formulation (F5-SA gel) with an SA to Arg·HCl ratio of 1:6, which significantly reduces rectal mucosal irritation while enhancing anti-inflammatory effects. Molecular docking revealed strong interactions between SA and Arg·HCl, suggesting reduced irritation. Physical evaluations confirmed favorable pH, stability, viscosity, and spreadability, making the gel suitable for rectal use. The anti-hemorrhoidal activity of the F5-SA gel was confirmed in animal models, where it promoted anal ulcer healing and effectively reduced inflammation and edema with minimal irritation. Safety assessments showed no significant toxicity, aside from a transient reduction in food intake immediately after administration. Additionally, no dermal sensitivity was observed, supporting its safety for topical use. These findings highlight the F5-SA gel as a promising therapeutic option for rectal conditions such as hemorrhoids, offering a more effective and safer alternative to traditional SA formulations. Future clinical studies are warranted to confirm these findings and further evaluate its potential for human use.

## Figures and Tables

**Figure 1 pharmaceutics-16-01498-f001:**
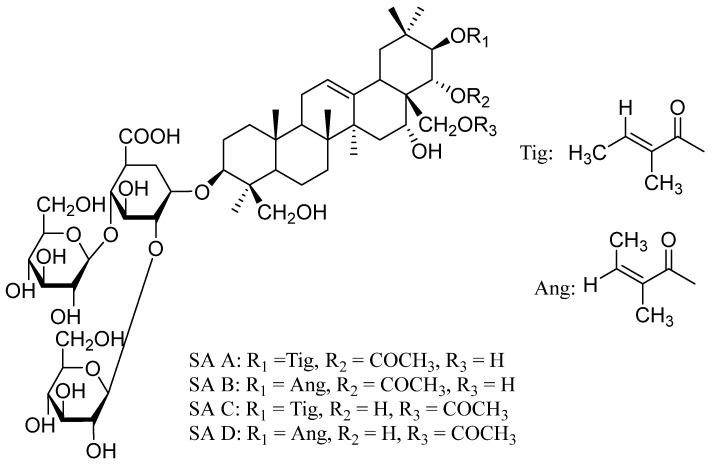
Chemical structures of different geometrical isomers of SA.

**Figure 2 pharmaceutics-16-01498-f002:**
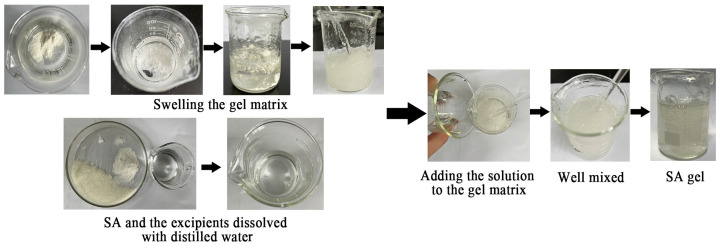
The procedure of the SA gel formulation.

**Figure 3 pharmaceutics-16-01498-f003:**
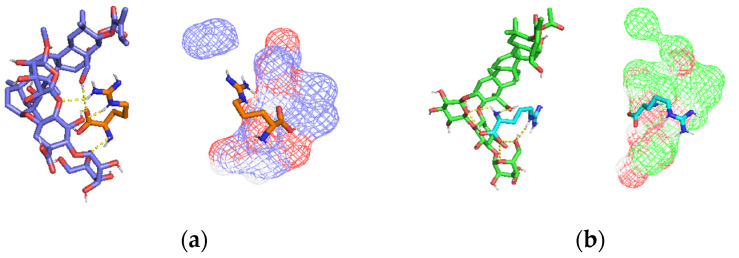
Predicted potential interactions between SA and Arg·HCl, showing an electrostatic map of SA’s surface, highlighting major areas of interaction. (**a**) SA A (skyblue: carbon atom, red: oxygen atom) and Arg·HCl (orange: carbon atom, blue: nitrogen atom); (**b**) SA B (green: carbon atom, red: oxygen atom) and Arg·HCl (cyan: carbon atom, blue: nitrogen atom); (**c**) SA C (yellow: carbon atom, red: oxygen atom) and Arg·HCl (magenta: carbon atom, blue: nitrogen atom); (**d**) SA D (pink: carbon atom and red: oxygen atom) and Arg·HCl (gray: carbon atom, blue: nitrogen atom).

**Figure 4 pharmaceutics-16-01498-f004:**
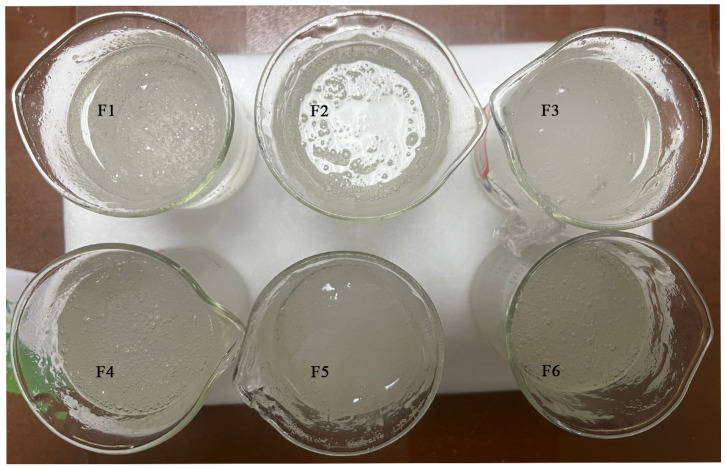
The organoleptic characteristics of the SA gels. F1~F6: six formulations with 2% SA and varying Arg·HCl ratios (SA:Arg·HCl = 1:0, 1:0.5, 1:1, 1:3, 1:6, and 1:10).

**Figure 5 pharmaceutics-16-01498-f005:**
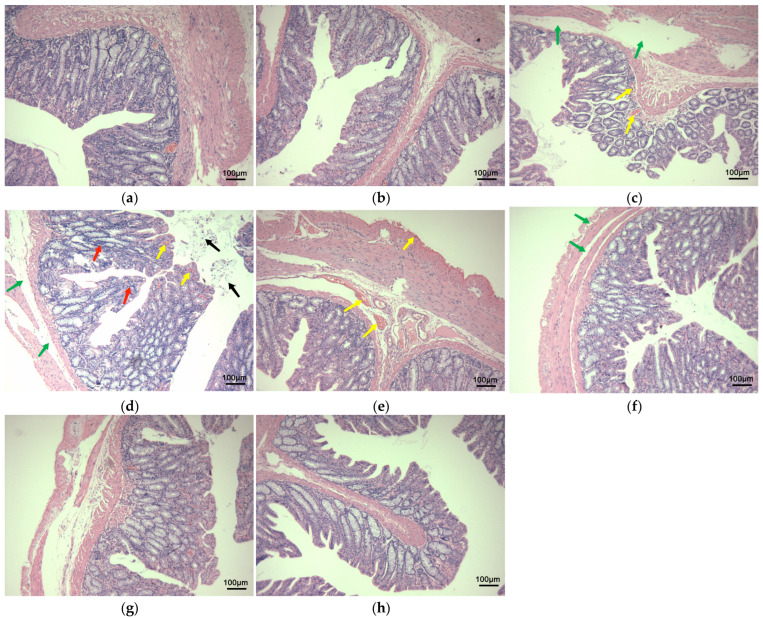
Histological sections of rectal mucosal stimulation for F1~F6 (HE staining ×100). (**a**) NS group; (**b**) SA gel matrix control group; (**c**) F1 group; (**d**) F2 group; (**e**) F3 group; (**f**) F4 group; (**g**) F5 group; (**h**) F6 group. Yellow arrows: hyperaemia; green arrows: mucosal surface shedding; red arrows: lymphocyte infiltration; black arrows: secretions.

**Figure 6 pharmaceutics-16-01498-f006:**
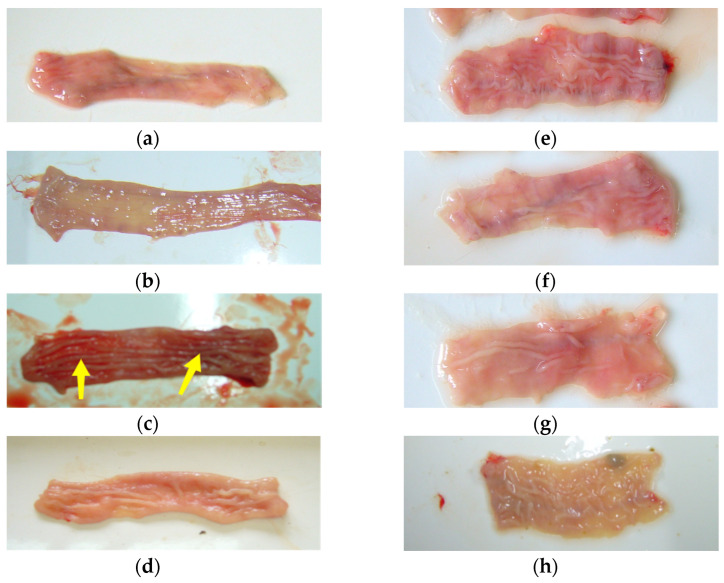
Gross observation of rectal mucosa one week after discontinuation of treatment for F1~F6. (**a**) NS group; (**b**) SA gel matrix control group; (**c**) F1 group; (**d**) F2 group; (**e**) F3 group; (**f**) F4 group; (**g**) F5 group; (**h**) F6 group. Yellow arrows denote hyperemia.

**Figure 7 pharmaceutics-16-01498-f007:**
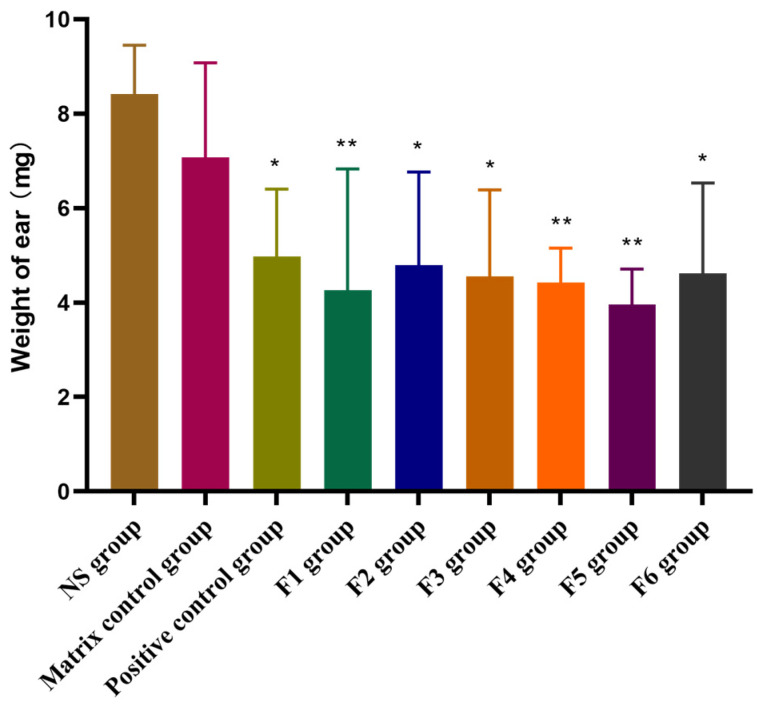
Effects of F1~F6 SA gels on xylene-induced ear swelling. Results are presented as mean ± SD (n = 10). One-way ANOVA followed by Dunnett’s test where * *p* < 0.05, ** *p* < 0.01, in comparison to matrix control group. (* moderately significant, ** highly significant).

**Figure 8 pharmaceutics-16-01498-f008:**
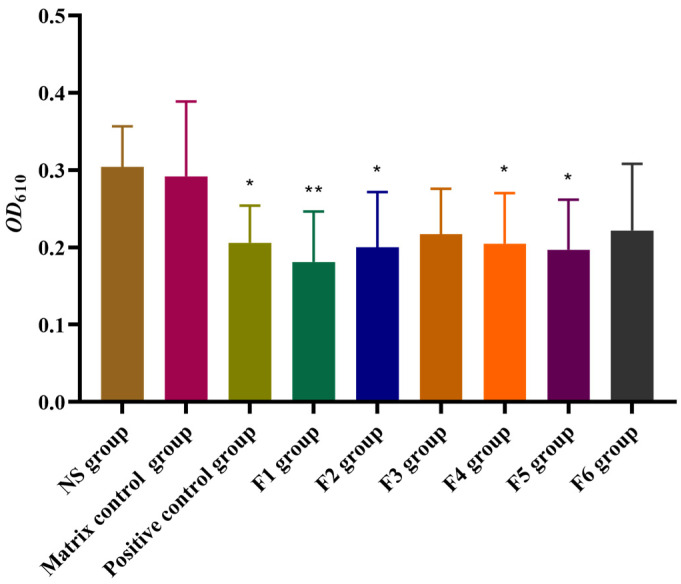
Effects of F1~F6 SA gels on histamine-induced capillary permeability. Results are presented as mean ± SD (n = 10). One-way ANOVA followed by Dunnett’s test where * *p* < 0.05, ** *p* < 0.01, in comparison to matrix control group. (* moderately significant, ** highly significant).

**Figure 9 pharmaceutics-16-01498-f009:**
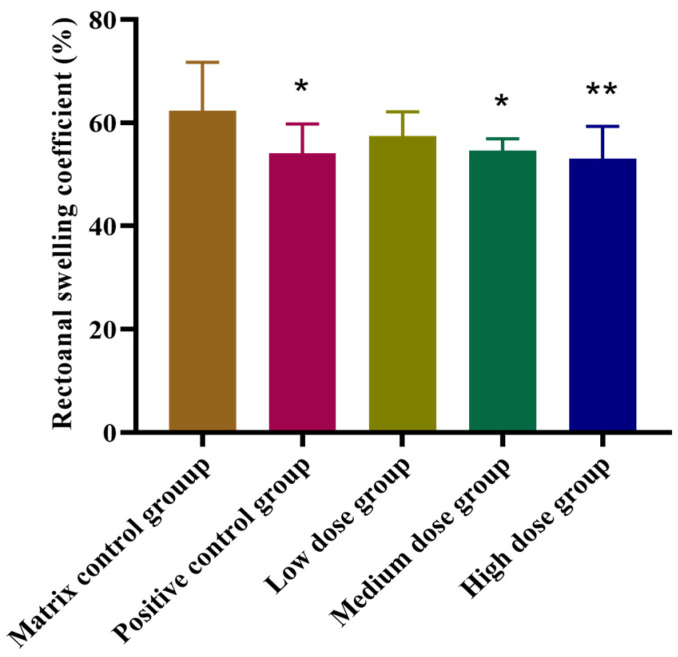
Effects of F5-SA gel on rectoanal swelling coefficient of hemorrhoidal rats induced by croton oil preparation. Results are presented as mean ± SD (n = 10). One-way ANOVA followed by Dunnett’s test where * *p* < 0.05, ** *p* < 0.01, in comparison to matrix control group. (* moderately significant, ** highly significant).

**Figure 10 pharmaceutics-16-01498-f010:**
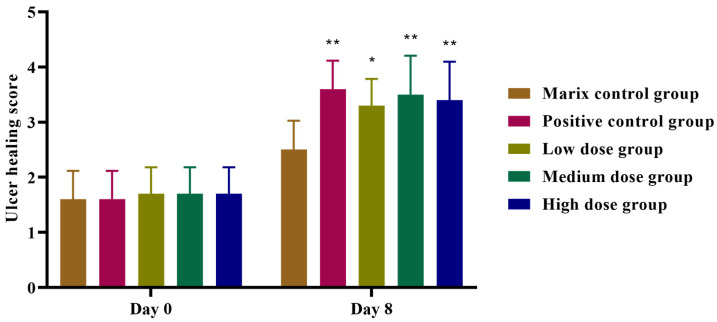
Effects of F5-SA gel on ulcer healing score of hemorrhoidal rabbits induced by acetic acid. Results are presented as mean ± SD (n = 10). One-way ANOVA followed by Dunnett’s test where * *p* < 0.05, ** *p* < 0.01, in comparison to matrix control group. (* moderately significant, ** highly significant). Day 0: before administration; Day 8: after 7 days of continuous administration.

**Table 1 pharmaceutics-16-01498-t001:** Formulations of SA gel.

Ingredients	F1	F2	F3	F4	F5	F6
SA (g)	2	2	2	2	2	2
Arg·HCl (g)	0	1	2	6	12	20
CMC-Na (g)	2.0	2.0	2.0	2.0	2.0	2.0
HPMC (g)	0.5	0.5	0.5	0.5	0.5	0.5
Glycerol (g)	10	10	10	10	10	10
Propylene glycol (g)	5	5	5	5	5	5
Edetate disodium (g)	0.01	0.01	0.01	0.01	0.01	0.01
Ethylparaben (g)	0.1	0.1	0.1	0.1	0.1	0.1
Water (g)	q.s 100 g					

**Table 2 pharmaceutics-16-01498-t002:** Physical properties evaluation of the SA gels.

Parameters	F1	F2	F3	F4	F5	F6
pH	7.00 ± 0.08	6.85 ± 0.05	6.56 ± 0.10	6.16 ± 0.06	6.00 ± 0.07	5.85 ± 0.03
Viscosity (cps)	378.4 ± 0.65	383.8 ± 0.36	390.8 ± 0.92	402.3 ± 0.86	411.2 ± 0.45	422.5 ± 0.63
Spreadability (cm)	4.37 ± 0.26	4.03 ± 0.36	3.84 ± 0.20	3.71 ± 0.30	3.50 ± 0.25	3.30 ± 0.15

**Table 3 pharmaceutics-16-01498-t003:** Body weight and mortality of the rabbits in acute toxicity test.

Weight	Group I	Group II
Initial weight (kg)	2.356 ± 0.076	2.310 ± 0.092
Day 1 weight (kg)	2.373 ± 0.036	2.310 ± 0.088
Day 2 weight (kg)	2.413 ± 0.033	2.338 ± 0.114
Day 5 weight (kg)	2.465 ± 0.038	2.413 ± 0.154
Day 7 weight (kg)	2.457 ± 0.047	2.432 ± 0.140
Day 10 weight (kg)	2.566 ± 0.051	2.516 ± 0.140
Day 14 weight (kg)	2.606 ± 0.076	2.614 ± 0.154
Body weight gain (%)	10.61	13.16
Mortality	No	No

Data are presented as mean ± SD (n = 6). Student’s *t*-test was used where no significant differences were found between the control and treatment groups.

**Table 4 pharmaceutics-16-01498-t004:** Food intake of the rabbits in acute toxicity test.

Food Intake	Group I (g/kg)	Group II (g/kg)
Initial day	64.1 ± 1.3	64.3 ± 3.0
Day 0	62.0 ± 3.0	46.2 ± 14.7 *
Day 1	62.8 ± 1.0	59.8 ± 9.8
Day 2	59.1 ± 4.1	58.6 ± 13.1
Day 5	59.7 ± 0.8	54.8 ± 14.4
Day 7	60.6 ± 1.3	57.5 ± 7.8
Day 10	57.7 ± 2.9	57.8 ± 2.7
Day 14	58.2 ± 1.9	57.2 ± 2.3

Data are presented as mean ± SD (n = 6). Student’s *t*-test was used where * *p* < 0.05 versus group I (* moderately significant).

**Table 5 pharmaceutics-16-01498-t005:** Body weight of dermal sensitivity testing in the guinea pig.

Group	No (n)	Before the Experiment	Last Sensitization	After the Experiment	Body Weight Gain
Matrix control (g)	10	335 ± 29	439 ± 28	522 ± 46	188 ± 42
Positive control (g)	10	329 ± 20	431 ± 18	504 ± 36	175 ± 24
F5-SA gel (g)	10	330 ± 25	427 ± 40	512 ± 63	182 ± 49

Data are presented as mean ± SD (n = 10). Student’s *t*-test was used where no significant differences were found vs. the matrix control.

**Table 6 pharmaceutics-16-01498-t006:** Guinea pig maximization testing of F5-SA gel.

Group	No (n)	Average Score				Animals with Allergic Reactions (n)				Sensitization Rate (%)			
		1 h	24 h	48 h	72 h	1 h	24 h	48 h	72 h	1 h	24 h	48 h	72 h
Matrix control	10	0	0	0	0	0	0	0	0	0	0	0	0
Positive control	10	3.8	2.8	1.5	0.6	10	10	10	6	100	100	100	60
F5-SA gel	10	0	0	0	0	0	0	0	0	0	0	0	0

## Data Availability

The data can be shared upon request.

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
