# Peer review of "Arginine Hydrochloride Reduce Rectal Mucosal Irritation of Sodium Aescinate: Molecular Docking, Physical Properties, Anti-Hemorrhoidal Activity, Safety and Topical Gel Formulations Investigation"

_pharmaceutics, 2024, doi:10.3390/pharmaceutics16121498_

Round 1

Reviewer 1 Report

Comments and Suggestions for Authors

This is an interesting study where products are formulated for the treatment of hemorrhoids or similar diseases. Apart from the physicochemical characterization that seems ideal to me, it carries out in vivo studies, which generates scientific solidity. However, there are some doubts that I would like to dispel before publishing this article.

1. Because they used the following animals: Kunming mice (18–22 g), SD rats (180 – 250 g) and Japanese big-eared rabbits (2.0 – 2.5 83 kilos). Could only one type of animal be used for the experiments or was it necessary to add mice and rabbits?

2. Could you add an image showing the procedure in section 2.3 of the gel formulation?

3. Why did they use parabens as a preservative?

4. Add an image of the organoleptic characteristics of the gels.

5. In extensibility, why was a weight of 100 g applied?

6. I don't understand why they carry out the xylol test if it is done in the mouse's ear. Does it have nothing to do with anal irritation? could you explain

7. In Figure 3 of the histology, place the affected and improved parts with arrows.

Reviewer 2 Report

Comments and Suggestions for Authors

Dear Authors, 

Congratulations on your work. Here are my comments: 

The Intorduction and the Conclusion are clear and consistent with the text. The References up to date, no inappropriate citations detected. The number of the cited references is appropriate for a review paper. The format of References are consistent. Overall, the Figures can be clearly understood and the Figure legends provide good explanation, the Figures have good appearance and resolution

For the better understandig please indicate (for example with arrows) the damaged histological structures on Figure 3 and Figure 4 

The viscosity values what you measured is appropriate for the human application? Is there any connection between the effectiveness of F5 group and viscosity value? 

How do you explain the increased viscosity with increasing concentration of Arginin? How the Arginin possibly increase the viscosity? 

Figure 8. Typo was detected : " control groupl"

How do you explain that F5 group is better in each experiment than F4 or F6? I miss the detailed explanation at each experiments, why F5 is the best? Do you have any supporting data? Please write at least 1-2  sentence explanation at each experiment and explain why F5 resulted the best.  
